# Toward Reliable Uptake Metrics in Large Vessel Vasculitis Studies

**DOI:** 10.3390/diagnostics11111986

**Published:** 2021-10-26

**Authors:** Gijs D. van Praagh, Pieter H. Nienhuis, Daniel M. de Jong, Melanie Reijrink, Kornelis S. M. van der Geest, Elisabeth Brouwer, Andor W. J. M. Glaudemans, Bhanu Sinha, Antoon T. M. Willemsen, Riemer H. J. A. Slart

**Affiliations:** 1Department of Nuclear Medicine and Molecular Imaging, University Medical Center Groningen, University of Groningen, 9700 RB Groningen, The Netherlands; p.h.nienhuis@umcg.nl (P.H.N.); daniel.m.dejong@gmail.com (D.M.d.J.); a.w.j.m.glaudemans@umcg.nl (A.W.J.M.G.); a.t.m.willemsen@umcg.nl (A.T.M.W.); r.h.j.a.slart@umcg.nl (R.H.J.A.S.); 2Division of Vascular Medicine, Department of Internal Medicine, University Medical Center Groningen, University of Groningen, 9700 RB Groningen, The Netherlands; m.reijrink@umcg.nl; 3Department of Rheumatology and Clinical Immunology, University Medical Center Groningen, University of Groningen, 9700 RB Groningen, The Netherlands; k.s.m.van.der.geest@umcg.nl (K.S.M.v.d.G.); e.brouwer@umcg.nl (E.B.); 4Department of Medical Microbiology and Infection Prevention, University Medical Center Groningen, 9700 RB Groningen, The Netherlands; b.sinha@umcg.nl; 5Department of Biomedical Photonic Imaging, University of Twente, 7500 AE Enschede, The Netherlands

**Keywords:** vasculitis, fluorodeoxyglucose F18, positron emission tomography computed tomography, standardized uptake values, lean body mass

## Abstract

The aim of this study is to investigate the influence of sex, age, fat mass, fasting blood glucose level (FBGL), and estimated glomerular filtration rate (eGFR) on blood pool activity in patients with large vessel vasculitis (LVV). Blood pool activity was measured in the superior caval vein using mean, maximum, and peak standardized uptake values corrected for body weight (SUVs) and lean body mass (SULs) in 41 fluorodeoxyglucose positron emission tomography/computed tomography (FDG-PET/CT) scans of LVV patients. Sex influence on the blood pool activity was assessed with t-tests, while linear correlation analyses were used for age, fat mass, FBGL, and eGFR. Significantly higher SUVs were found in women compared with men, whereas SULs were similar between sexes. In addition, higher fat mass was associated with increased SUVs (r = 0.56 to 0.65; all *p* < 0.001) in the blood pool, but no correlations were found between SULs and fat mass (r = −0.25 to −0.15; all *p* > 0.05). Lower eGFR was associated with a higher FDG blood pool activity for all uptake values. In FDG-PET/CT studies with LVV patients, we recommend using SUL over SUV, while caution is advised in interpreting SUV and SUL measures when patients have impaired kidney function.

## 1. Introduction

Semi-quantitative measurements, e.g., the mean or maximum standardized uptake value (SUV), are increasingly being used and recommended in 2-[fluorine-18]-fluoro-2-deoxy-D-glucose positron emission tomography (FDG-PET). The SUV is a simple, unitless metric in which the amount of activity within a volume of interest (VOI) in the PET scan is corrected for the injected radiotracer dose and for body weight. In 1993, Zasadny et al. recommended to use the SUV normalized to lean body mass (LBM), referred to as SUL, due to the SUV’s dependency on body weight [1]. FDG uptake in fat is low, whereas body weight highly depends on fat [2]. This would imply that correcting for body weight is inadequate and might lead to different treatment approaches between patients with different fat percentages or to erroneous patient monitoring when fat mass changes. Thus, using SUL should result in more consistent quantification of the resulting signal. SUV and SUL are calculated as follows:SUV or SUL = ActVOIkBqmLActadministeredMBqBW or LBMkg,
where *Act_VOI_* is the activity concentration measured in the VOI; the *Act_administered_* is the net administered activity corrected for physical decay of FDG to the start of the acquisition; and 1 mL of tissue is assumed to weigh 1 g [3].

The most recent EANM guideline for oncological imaging recommends quantifying the tracer uptake by using SUL [3]. However, in PET imaging of inflammatory disorders, such as large vessel vasculitis (LVV), there is no consensus on the best correction method of FDG uptake [4,5]. Most (recent) LVV PET studies only used SUV as quantitative scoring method and did not mention SUL [6,7,8,9].

Furthermore, recent literature demonstrated differences in glucose metabolism with sex and age, possibly influencing the uptake activity during scanning [10]. Besides, impaired renal function slows down the clearance of FDG and thus may increase the blood pool activity at the standard 60 min acquisition time [11]. Guidelines consider this issue when intravenous contrast material has to be applied, but do not advise for FDG [3,4].

Although these semi-quantitative measurements are mainly used in research settings of patients with LVV so far, it is of utmost importance that these values are correct and reliable to potentially use FDG PET to achieve optimal diagnosis, therapy monitoring, and consistent treatment strategies. Therefore, the aim of this study was to investigate the influence of sex, age, fat mass, fasting blood glucose level (FBGL), and kidney function on blood pool SUVs and SULs in patients with large vessel vasculitis.

## 2. Materials and Methods

### 2.1. Patients and Scan Acquisition

In this study, electronic patient files of patients with LVV were retrospectively checked for the presence of a FDG PET/CT scan before start of treatment. All patients signed informed consent as part of a prospective cohort study, which was approved by the institutional review board of the UMCG (METc 2010/222) [12]. The LVV diagnosis was made by a rheumatologist. This diagnosis had to be manifested for at least 6 months after initial presentation.

All scans were performed using an integrated PET/CT system (Biograph mCT 40 or 64-slice or Vision; Siemens, Knoxville, TN, USA). The FDG-PET/CT procedure was performed according to the EANM guidelines, which included at least 6 h fasting prior to FDG injection (3 MBq/kg), subsequent 60 min waiting, and image acquisition of 3 min per bed position. All scans were reconstructed according to EARL for semi-quantitative analysis [13].

### 2.2. Quantification

Blood pool activity was measured by drawing VOIs within the boundaries of the superior caval vein (SCV) using Hermes Affinity Viewer v2.02 software (Hermes Medical Solutions Inc., Greenville, NC, USA). The SCV was manually delineated on all low-dose CT slices where the SCV was visible. Manual delineation started cranially at the lower border of the first right costal cartilage (where the left and right brachiocephalic veins end). The SCV was delineated until the first CT slice where the SCV was not distinguishable anymore from the right atrium. After overlaying the co-registered PET image with the low-dose CT, voxels with FDG-uptake spillover from neighboring tissue (i.e., heart tissue or the aortic wall) were carefully excluded from the SCV VOI (see Figure 1 for an example). Delineation was performed by an MD/PhD student with three years of experience with vessel segmentation in PET/CT. The SCV was segmented a second time in a random sample of 10 patients (25%) in order to assess intra-observer reliability.

The mean, peak, and maximum standardized uptake value (SUV_mean_ and SUL_mean_, SUV_peak_ and SUL_peak_, and SUV_max_ and SUL_max_) of the SCV were chosen as activity parameters. LBM was calculated according to Janmahasatian et al. [14] for male and female, respectively, as recommended in the EANM guidelines for oncological imaging [3]:$$LBM^{M}\;=\;9270\;\times \;\frac{body\;weight}{6680\;+\;216\;\times \;BMI},$$
$$LBM^{F}\;=\;9270\;\times \;\frac{body\;weight}{8780\;+\;244\;\times \;BMI}.$$

### 2.3. Statistical Analysis

Correlation analyses were performed comparing the activity parameters in the SCV according to age, FBGL (mmol/L), fat mass (= body weight − LBM), and estimated glomerular filtration rate (eGFR) calculated using the Chronic Kidney Disease Epidemiology Collaboration (CKD-EPI) equation [15]. Pearson’s r was calculated when activity parameters and possible influencing factors were normally distributed. Spearman’s r was calculated when activity parameters and possible influencing factors were not normally distributed. Student’s t-test or Mann-Whitney U test was done to compare groups (i.e., male vs. female) after testing for normality. After Holm-Bonferroni correction, results were considered statistically significant when *p* < 0.05. An intraclass correlation coefficient (two-way random model) was calculated to assess the intra-observer reliability. Coefficients between 0.75 and 1.00 were considered excellent, between 0.60 and 0.74 good, between 0.40 and 0.59 fair, and below 0.40 poor. Statistical tests were performed in Graphpad Prism 8 (GraphPad Software, San Diego, CA, USA) or SPSS (IBM Corp. Released 2015. IBM SPSS Statistics for Windows, Version 23.0. Armonk, NY: IBM Corp.).

## 3. Results

Patients were selected from a prospective cohort with newly diagnosed LVV patients. In total, 43 patients from this cohort underwent an FDG-PET/CT scan at time of diagnosis before start of treatment between 2011 and 2020, and were included in this study. Two patients were excluded because of missing eGFR and height data. Patient characteristics are presented in Table 1. Patient characteristics were equal between female and male patients, except for LBM and fat mass, which were, respectively, lower and higher in females than males.

First, we investigated whether the patient demographics of sex and age influenced the blood pool activity (Figure 2). Blood pool activity measured as SUV_mean_, SUV_peak_, and SUV_max_ was significantly higher in females compared with males (*p* = 0.016, *p* = 0.010, and *p* = 0.010, respectively). However, when measured as SUL_mean_, SUL_peak_, and SUL_max_, no significant differences were found (*p* = 0.087, *p* = 0.195, and *p* = 0.114, respectively). No association between age and blood pool activity was found (Table 2 and Figure 3).

Increased fat mass, determined by subtracting the LBM from the total body mass, was associated with increased SUV_mean_, SUV_peak_, and SUV_max_ in the blood pool (Figure 4). When correcting for lean body mass by using SUL, no significant correlations were found with fat mass (Table 2)

FBGL was not associated with FDG activity in the blood pool (Table 2 and Figure 5). Conversely, decreased kidney function (in eGFR) was associated with a higher FDG activity in the blood pool (Table 2 and Figure 6). Numerical presentation of the uptake metrics shown in the graphs may be found in the Appendix A.

Intraclass correlation coefficients of two separate measurements by the same observer yielded excellent or good coefficients of 0.86 for SUV_mean_, 0.71 for SUV_peak_, 0.81 for SUV_max_, 0.88 for SUL_mean_, 0.69 for SUL_peak_, and 0.83 for SUL_max_.

## 4. Discussion

Guidelines for [18F]FDG-PET/CT imaging in oncological diseases recommend the use of SUL for semi-quantitative analysis instead of the commonly used SUV [3]. In inflammatory diseases there is no consensus regarding which parameter to use for analysis. Therefore, we investigated different quantitative measurements (SUV and SUL) for defining the blood pool activity in a specific patient group with LVV. Besides a higher fat mass in female patients compared with male patients, the results showed a significantly higher SUV_mean_, SUV_max_, and SUV_peak_ in women compared with men. When corrected for LBM, no significant differences were found (for SUL_mean_, SUL_max_, and SUL_peak_). Furthermore, a significant positive correlation between SUVs and fat mass was found, meaning that a higher fat mass may slow down or reduce the FDG uptake in organs and tissues, thereby subsequently increasing the blood pool FDG activity. To better understand the underlying mechanisms, exact interactions should be investigated in future research. When corrected for LBM (SUL), no correlation was noted. Additionally, no association was found between FBGL and blood pool activity, whereas a negatively proportional correlation was found between eGFR and blood pool activity, meaning that impaired kidney function may result in higher blood pool activity values through reduced renal FDG clearance.

To compare between patients and for consistent treatment strategies it is of utmost importance that imaging metrics correctly reflect the severity of the disease. The population variation of LVV patients, including GCA and Takayasu’s arteritis, is large [16]. Large differences in body weight and amount of body fat between and within patients over time may exist, implying that SUV measurements might lead to erroneous interpretation of the (follow-up) results. Our results, although with a small sample size, strongly suggest that the amount of body fat influences the blood pool activity, leading to differences between females and males in SUVs. This could be explained by the fact that, in general, females have a higher percentage of body fat relative to their body weight than men [17]. SUL could thus be a more stable metric, also supported by our findings. This is in line with previous research from Sarikaya et al. which found that SUV overestimates metabolic activity in all patients, but even more in obese patients [18]. SUL, however, is not affected by body fat, and thus more accurate.

Besides SUL, the target-to-background ratio (TBR) is also a metric that may be more reliable in some diseases, especially in therapy follow up studies, and is recommended by guidelines for LVV imaging using blood pool as background [4]. Because TBR is calculated by dividing the SUV or SUL in the target tissue by the SUV or SUL in the background, the correction methods are also divided by each other and thus do not play a role anymore:$$TBR=\frac{SUV\;or\;SUL\left(VOI\right)}{SUV\;or\;SUL\left(background\right)}.$$

Contradictory to previous research [19,20,21], we did not find any association between FBGL and blood pool activity. This could be explained by the low variance in FBGL of our patient dataset; over 80% of the patients had an FBGL level lower than 7.0 mmol/L, which was proposed as upper cut-off value whether to perform the [18F]FDG-PET scan at that moment by Bucerius et al. [19].

We did find a negative correlation between blood pool activity values and eGFR. This is in line with results from previous studies. Rosenblum et al. showed in both Takayasu’s and GCA patients a negative correlation between blood pool activity and GFR during 1-h imaging [22]. Derlin et al. demonstrated in 50 [18F]FDG-PET scans a significant negative correlation between blood pool SUVs and eGFR [23]. Both studies warn that this could result in overcorrection when TBRs are calculated, as GFR was not associated with FDG uptake in the arterial wall [22,23]. In the same study, Rosenblum et al. showed that at 2-h imaging blood pool activity and TBRs were not associated with GFR anymore [22]. Furthermore, Toriihara et al. found significantly higher uptake values in soft tissues, spleen, and blood pool in chronic renal failure patients compared with the control group with normal kidney function [24]. Laffon et al. even proposed to use a lower [18F]FDG radiation dose in patients with renal failure by using their two-compartment model, which would result in similar uptake values due to the decreased clearance [25]. As stated before, current guidelines do not address this topic concerning the use of [18F]FDG. Some of the above-mentioned problems may be eliminated by dynamic or dual time point PET, which should be investigated for LVV patients in future research [26,27]. However, as this is currently not clinical practice yet, based on our results, we suggest being cautious with interpreting uptake measurements when patients have impaired renal function.

Our study has some limitations. First, the number of patients included was relatively small. To make recommendations for specific cutoff values in renal function or lower administration of radiotracer dose for example, a larger sample size would be necessary. However, our results coincide with previous research in other diseases and therefore show the importance of consistent scoring in LVV. Second, the variance in FBGL was small and levels itself were mostly low compared with previous proposed cutoff values. Last, only one author delineated the SCVs and therefore interobserver variability analysis was not done. However, intra-observer reliability was rated excellent or good.

In conclusion, fat mass has a significant influence on blood pool SUVs, resulting in differences between sexes. SUL scores are not dependent of the amount of body fat in a patient and thus result in more realistic scores and potentially more consistent treatment strategies than SUV measurements. In addition, eGFR influences blood pool activity in a negative proportional manner. Therefore, we recommend using SUL instead of SUV as quantitative measurement in [18F]FDG-PET scans of patients with LVV and to be cautious with the interpretation of quantitative measurements when patients have impaired renal function.

## Figures and Tables

**Figure 1 diagnostics-11-01986-f001:**
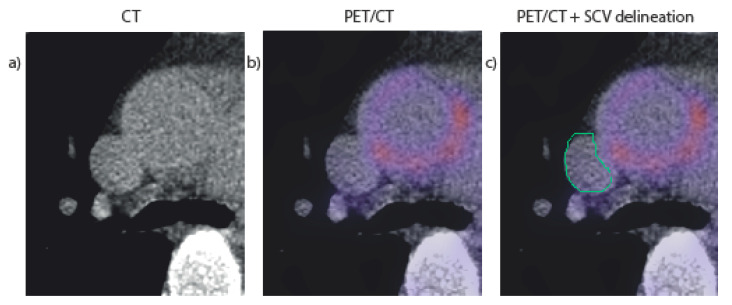
Axial slice of the superior caval vein (SCV) on a CT only (**a**), on the CT and PET fused (**b**), and the PET/CT with the SCV manually delineated, while carefully excluded the spillover from the heart next to it (**c**).

**Figure 2 diagnostics-11-01986-f002:**
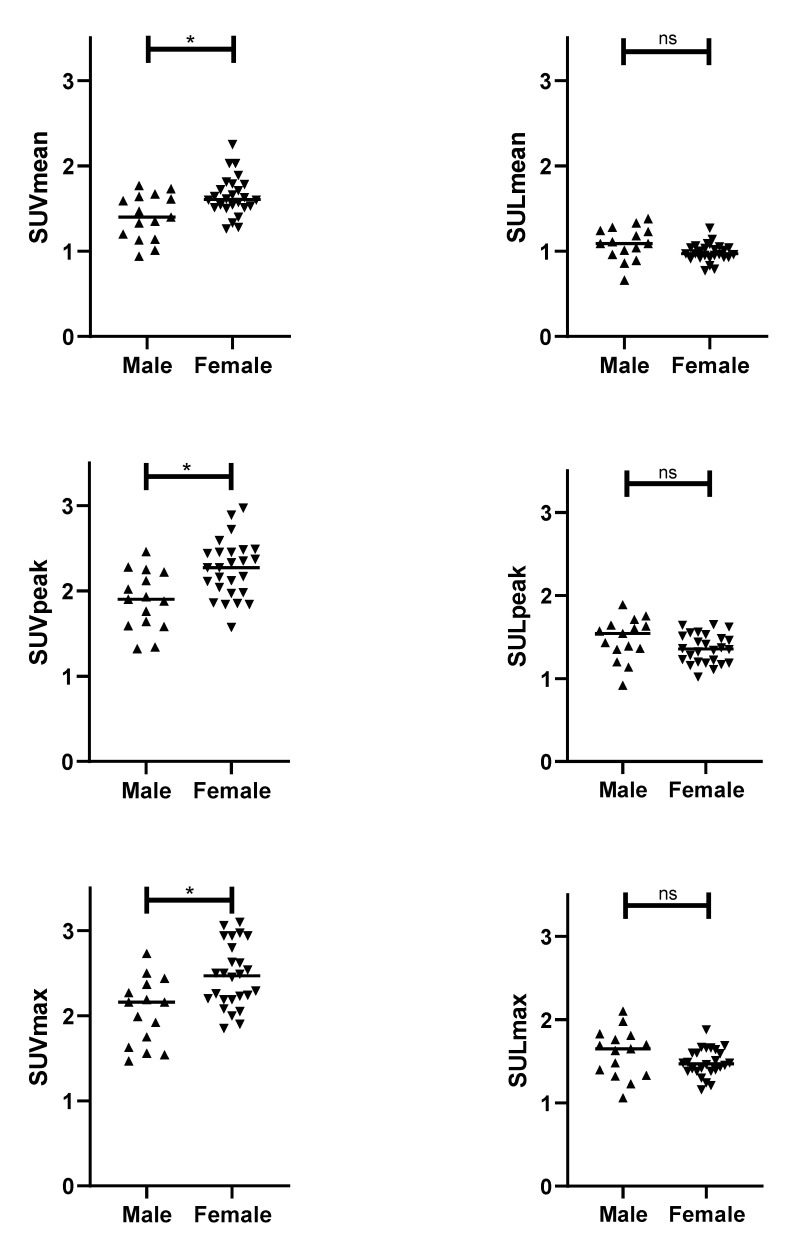
Sex differences in blood pool activity as measured by the mean standardized uptake value (SUV_mean_), peak standardized uptake value (SUV_peak_), maximum standardized uptake value (SUV_max_), mean standardized uptake value corrected for lean body mass (SUL_mean_), peak standardized uptake value corrected for lean body mass (SUL_peak_), and maximum standardized uptake value corrected for lean body mass (SUL_max_). Results from t-tests are shown in the graphs, with * = *p* < 0.05 and ns = not significant.

**Figure 3 diagnostics-11-01986-f003:**
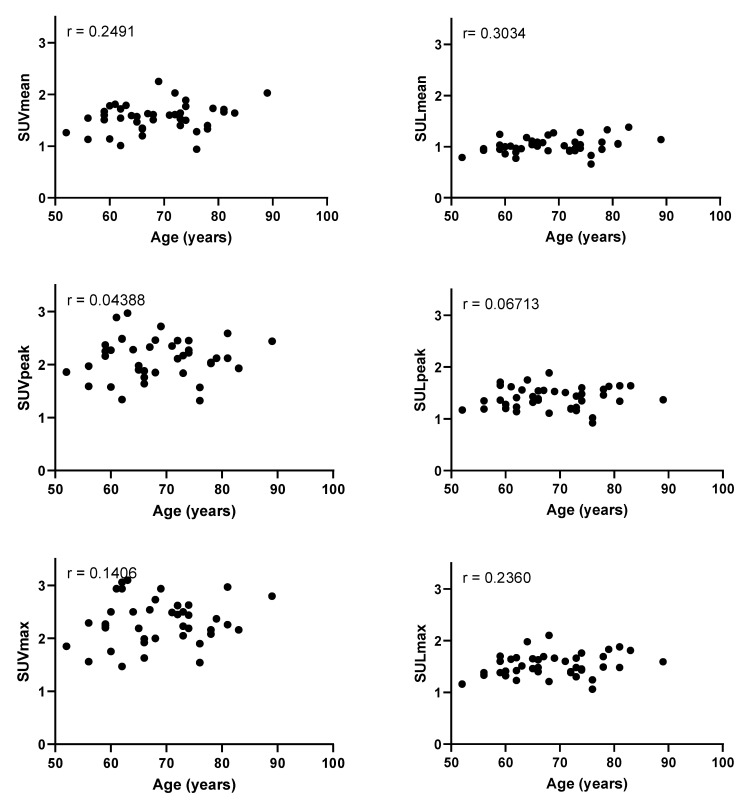
Plots of the patients’ ages in years and blood pool activity as measured by the mean standardized uptake value (SUV_mean_), peak standardized uptake value (SUV_peak_), maximum standardized uptake value (SUV_max_), mean standardized uptake value corrected for lean body mass (SUL_mean_), peak standardized uptake value corrected for lean body mass (SUL_peak_), and maximum standardized uptake value corrected for lean body mass (SUL_max_). Pearson’s r values are shown in the upper left corners. None of the correlations were statistically significant (*p* = 0.582, *p* = 0.785, *p* = 1.000, *p* = 0.323, *p* = 1.000, *p* = 0.550 for SUV_mean_, SUV_peak_, SUV_max_, SUL_mean_, SUL_peak_, SUL_max_, respectively).

**Figure 4 diagnostics-11-01986-f004:**
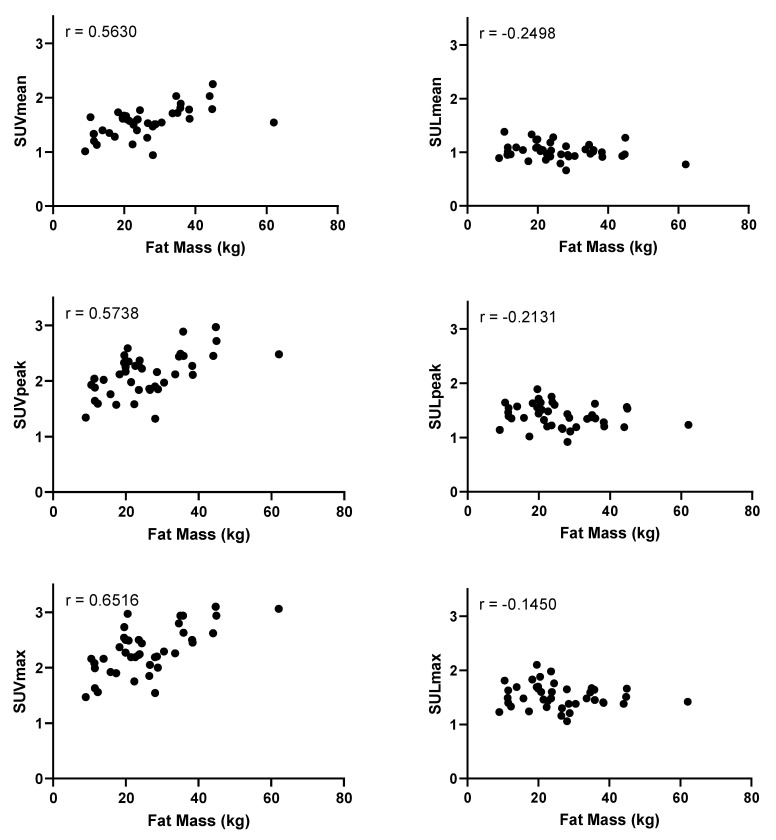
Plots of fat mass in kilograms and blood pool activity as measured by the mean standardized uptake value (SUV_mean_), peak standardized uptake value (SUV_peak_), maximum standardized uptake value (SUV_max_), mean standardized uptake value corrected for lean body mass (SUL_mean_), peak standardized uptake value corrected for lean body mass (SUL_peak_), and maximum standardized uptake value corrected for lean body mass (SUL_max_). Fat mass was calculated by subtracting the lean body mass from the total body weight. Pearson’s r values are shown in the upper left corners, with *p* < 0.001 (SUV_mean_), *p* < 0.0001 (SUV_peak_), *p* < 0.0001 (SUV_max_), and no statistical significance for SUL_mean_ (*p* = 0.346), SUL_peak_ (*p* = 0.362), and SUL_max_ (*p* = 0.366).

**Figure 5 diagnostics-11-01986-f005:**
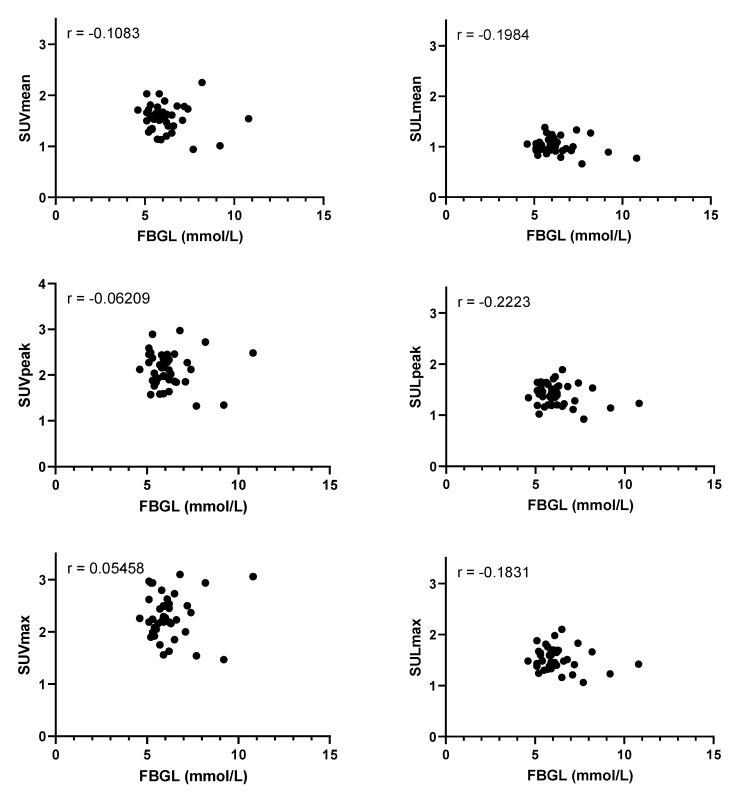
Plots of fasting blood glucose level (FBGL) in mmol/L and blood pool activity as measured by the mean standardized uptake value (SUV_mean_), peak standardized uptake value (SUV_peak_), maximum standardized uptake value (SUV_max_), mean standardized uptake value corrected for lean body mass (SUL_mean_), peak standardized uptake value corrected for lean body mass (SUL_peak_), and maximum standardized uptake value corrected for lean body mass (SUL_max_). Pearson’s r values are shown in the upper left corners. None of the correlations were statistically significant (*p* = 1.000, *p* = 1.000, *p* = 0.735, *p* = 1.000, *p* = 0.974, *p* = 1.000 for SUV_mean_, SUV_peak_, SUV_max_, SUL_mean_, SUL_peak_, SUL_max_, respectively).

**Figure 6 diagnostics-11-01986-f006:**
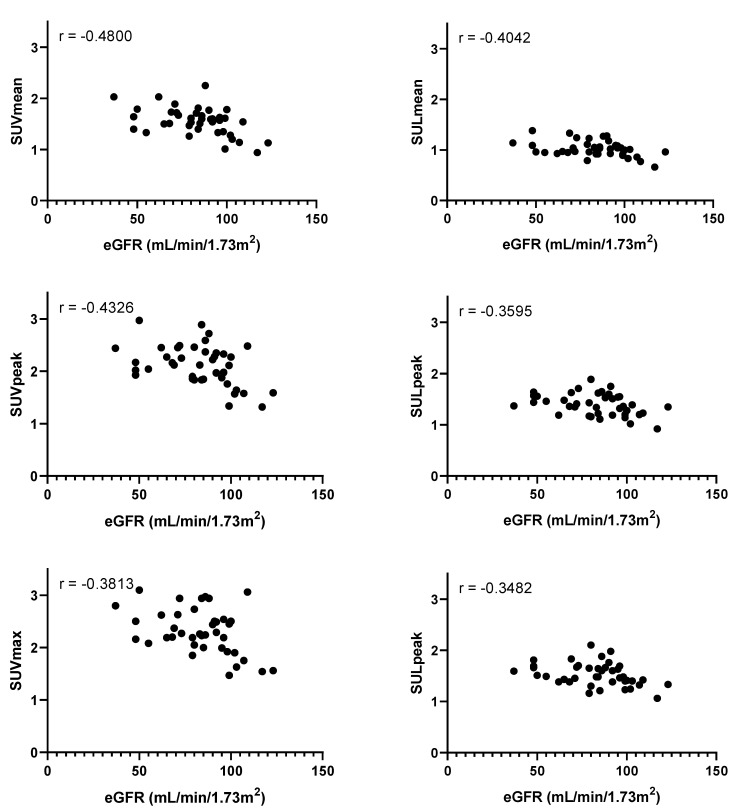
Plots of the estimated glomerular filtration rate (eGFR) and blood pool activity as measured by the mean standardized uptake value (SUV_mean_), peak standardized uptake value (SUV_peak_), maximum standardized uptake value (SUV_max_), mean standardized uptake value corrected for lean body mass (SUL_mean_), peak standardized uptake value corrected for lean body mass (SUL_peak_), and maximum standardized uptake value corrected for lean body mass (SUL_max_). Pearson’s r values are shown in the upper left corners, with *p* = 0.009 (SUV_mean_), *p* = 0.035 (SUL_mean_), *p* = 0.024 (SUV_peak_), *p* = 0.042 (SUL_peak_), *p* = 0.042 (SUV_max_), *p* = 0.026 (SUL_max_).

**Table 1 diagnostics-11-01986-t001:** Patient characteristics of all patients together and female and male separately. Statistically significant *p*-values from comparisons between females and males are highlighted in bold.

	All Patients			Female			Male			F vs. M
Patient Characteristics	n [%]	Mean ± SD	Range	n [%]	Mean ± SD	Range	n [%]	Mean ± SD	Range	*p*-Value
Number of patients	41			26 [63.4]			15 [36.6]			
Age (years)		68.6 ± 8.4	52–89		68.9 ± 8.7	52–83		68.1 ± 8.1	56–83	0.7968
Weight (kg)		76.6 ± 16.1			75.5 ± 18.1			78.4 ± 12.1		0.5900
BMI (kg/m^2^)		26.0 ± 4.8			26.8 ± 5.3			24.5 ± 3.6		0.1354
LBM (kg)		50.8 ± 10.2			45.2 ± 7.4			60.5 ± 6.3		**<0.0001**
Fat mass (kg)		25.8 ± 11.3			30.4 ± 11.1			17.9 ± 6.4		**0.0003**
FBGL (mmol/L)		6.2 ± 1.2	4.6–10.8		6.1 ± 1.2	4.6–9.2		6.4 ± 1.0	5.3–9.2	0.2332
eGFR (mL/min/1.73 m^2^)		82.7 ± 19.9	37–123		79.6 ± 18.2	37–123		88.0 ± 22.1	48–123	0.1957
Glucocorticoid naive	31 [75.6]			20 [76.9]			11 [73.3]			

SD = standard deviation; BMI = body mass index; LBM = lean body mass; FBGL = fasting blood glucose level; eGFR = estimated glomerular filtration rate.

**Table 2 diagnostics-11-01986-t002:** Pearson’s correlation coefficients for age, fat mass, FBGL, and eGFR, with the PET activity parameters. The 95% confidence intervals are shown between brackets. Significance of the correlations is denoted by * (*p* < 0.05), ** (*p* < 0.01), *** (*p* < 0.001), and **** *p* < 0.0001).

	Age	Fat Mass	FBGL	eGFR
SUV_mean_	0.249 (−0.063 to 0.517)	0.563 *** (0.309 to 0.742)	−0.108 (−0.403 to 0.206)	−0.480 ** (−0.686 to −0.202)
SUL_mean_	0.303 (−0.005 to 0.559)	−0.250 (−0.518 to 0.063)	−0.198 (−0.477 to 0.116)	−0.404 * (−0.633 to −0.110)
SUV_peak_	0.044 (−0.267 to 0.347)	0.574 **** (0.323 to 0.749)	−0.062 (−0.363 to 0.250)	−0.433 * (−0.653 to −0.144)
SUL_peak_	0.067 (−0.246 to 0.367)	−0.213 (−0.489 to 0.101)	−0.222 (−0.496 to 0.092)	−0.360 * (−0.601 to −0.058)
SUV_max_	0.141 (−0.175 to 0.430)	0.652 **** (0.430 to 0.799)	0.055 (−0.257 to 0.356)	−0.381 * (−0.617 to −0.083)
SUL_max_	0.236 (−0.077 to 0.507)	−0.145 (−0.433 to 0.170)	−0.183 (−0.465 to 0.132)	−0.348 * (−0.592 to −0.045)

SUV = standardized uptake value corrected by body weight; SUL = standardized uptake value corrected by lean body mass; FBGL = fasting blood glucose level; eGFR = estimated glomerular filtration rate.

## Data Availability

The data are not publicly available due to the still ongoing prospective study.

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
