# Peer review of "Toward Reliable Uptake Metrics in Large Vessel Vasculitis Studies"

_diagnostics, 2021, doi:10.3390/diagnostics11111986_

Round 1

Reviewer 1 Report

The effect of GFR on blood pool activity and target activity should be explained more. In addition, dual-time point PET imaging should be discussed as a potential solution to overcome blood pool temporal change. You may find this article useful:

van den Hoff J, Hofheinz F, Oehme L, Schramm G, Langner J, Beuthien-Baumann B, et al. Dual time point based quantification of metabolic uptake rates in 18F-FDG PET. EJNMMI Res. 2013;3: 16. doi:10.1186/2191-219X-3-16

and also

Blomberg BA, Bashyam A, Ramachandran A, Gholami S, Houshmand S, Salavati A, et al. Quantifying [18F]fluorodeoxyglucose uptake in the arterial wall: the effects of dual time-point imaging and partial volume effect correction. Eur J Nucl Med Mol Imaging. 2015;42: 1414–1422. doi:10.1007/s00259-015-3074-x

The relative independence of TBR to GFR at 2hour PET is reflected in reference #22 (Rosenblum JS, Quinn KA, Rimland CA, Mehta NN, Ahlman MA, Grayson PC. Clinical Factors Associated with Time-Specific Distribution of 18F-Fluorodeoxyglucose in Large-Vessel Vasculitis. Sci Rep. 2019;9: 15180. doi:10.1038/s41598-019-51800-x)

Reviewer 2 Report

1.     General

There are a lot of abbreviations in the article, a table with all abbreviations and their explanations can be considered. You chose FBG as an abbreviation for fasting blood glucose, which is correct, but unfortunately, is very close to FDG. I therefore suggest using another abbreviation (for example FBS or FBGL).

I strongly recommend reading https://doi.org/10.1038/s41598-020-74443-9 and using the same abbreviations, where applicable. This article could also serve as an additional reference.

There is unfortunately a problem with references in the text, first time is on line 110. Check the whole document and adjust accordingly.

On line 27 and 175 is stated that “Higher SULmean were found in men” but on line 117, the opposite is stated “When blood pool activity was measured by SULmean, females showed higher uptake values than males (p=0.029).”

2.     Abstract

More precision is needed.  I recommend something like the following:

The aim of this study is to investigate the influence of sex, age, fat mass, fasting blood glucose (FBGL) and estimated glomerular filtration rate (eGFR) on the blood pool activity (BPa) in 41 large vessel vasculitis (LVV) patients, using fluorodeoxyglucose positron emission tomography/ computed tomography (FDG-PET/CT). BPa was measured in the superior caval vein using mean, maximum, and peak standardized uptake values (SUVmn, SUVmx, SUVpk) and the equivalent SUV metrics per lean body mass (SULmn, SULmx, SULpk). Sex influence on the BPa was assessed with t-test, while linear correlation analyses were used for age, FBGL, fat mass and eGFR. Women had significantly higher SUVs and SULmean values. Both fat mass and eGFR corelated positively and significantly with SUV values, this also being true for eGFR and SUL values. In FDG-PET/CT of LVV patients, we recommend using SUL over SUV, while extra caution is advised in interpreting SUV and SUL measures if eGFR is low.

3.     Materials and Methods

Start with cut and paste line 106-109. Then add methodology for eGFR, blood glucose and fat mass calculation. For the 2.2 section, I recommend being more specific. For example:

Blood pool activity was measured by drawing volumes of interest (VOI) within the boundaries of the superior caval vein (SCV) using Hermes Affinity Viewer v2.02 software (Hermes Medical Solutions Inc., Greenville, NC, 85 USA). The SCV was manually delineated on all low-dose CT slices, beginning cranially, on the last slice where right costal cartilage is visible, and ending caudally, where the third costal cartilage disappears.  After overlaying the PET image with the low-dose CT, voxels with FDG-uptake spillover from neighboring tissue (i.e., heart tissue or the aortic wall) were carefully excluded from the SCV VOI.

Also in the 2.2 section, I have a few questions:

1) who did the manual segmentation?

2) what about inter- and intra-observer variability? “Free-hand manual segmentations” are known to be prone to both.

3) How was the PET and CT fused and aligned? Rigid vs non-rigid, DICOM-based or software based?

4) Did you use a specific predefined PET/CT merging protocol mentioned in the literature, and if yes, reference please?

5) Could you show some screenshots?

  1. of an axial CT slice, without segmentation
  2. same slice, with CT segmentation
  3. same slice, with CT segmentation and PET overlap
  4. same slice, with FDG spillover activity removed
  5. all those 4 images could be a put in a row from left to right, and presented as a single figure.

4.     Results

Table 1 can be improved. I recommend the following layout:

Metric

n

mn

sd

min

max

Line 114-116 can be simplified to: females had a higher BP activity than males, in terms of SUVmean, SUVpeak, SUVmax and SULmean (…).

5.     Discussion

Very nicely written, and good to see TBR in perspective. I miss a few words on the method limitations (such as time, skill, precision, and repeatability) and considerations in this regard.

I would also like if you can make the following table, as appendix, with all SUV and SUL metrics for all patients in one group, including the actual size (in cubic centimeters) for the SCV VOI.

Metric

n

mn

sd

min

max

Vol

SUVmn

SUVmx

SUVpk

SULmn

SULmx

SULpk

Reviewer 3 Report

General comments:

This manuscript addresses important issues related to the validity of SUV as a semiquantitative measure of FDG uptake in PET scans of patients with large vessel vasculitis. It has a clear purpose, is well written and the results may influence future practice. I think; however, that there are areas in the text that need a more detailed description as well as questions regarding statistical analysis and interpretations that need answering before this manuscript is suited for publication.

Specific comments:

P2L72: How was the LVV diagnosis defined and obtained? Please elaborate.

P2L73: Were there any exclusion criteria?

P2L74: What is GPS? Please define.

P2L77: Two different scanner systems were used, and SUV´s can vary significantly between different scanners – is this not a possible confounder that should be checked?

P2L79: How was the FDG dose determined? Fixed or adjusted to weight?

P3L96: Many different measurements are compared with a lot different variables increasing the risk of spurious significance. Were sample size calculations performed? Alternatively, although conservative, a Bonferroni correction of the significance level would strengthen the claimed significant effects.

In order to better understand the significance and effect size of the various factors a more integrated statistical analysis is suggested e.g. generalized linear model or similar approach. In such models interactions e.g. between fat mass and gender can be tested as well.

P3L108 & Table 1: It is stated both that subjects were scanned before treatment start, and that only 31 % were glucocorticoid naïve. This needs clarification, as prior glucocorticoid treatment may affect blood pool activity.

P10L177-178: No time activity relationship exists in these data to support this hypothesis/claim.

P10L191: The authors claim that increased blood pool activity in women is due to higher amount of body fat and in figure 3 a positive correlation between absolute fat mass and SUV is shown. Even though women in general have higher body fat percentage, this does not mean that the absolute fat mass is higher – especially in a relatively small population. Furthermore, it is not clear in the text that women in the manuscript had a significantly higher total fat mass. Thus I am not convinced by the argument. An idea to support this argument is to further subdivide the plots in gender categories.

Round 2

Reviewer 3 Report

Thank you for a thorough and relevant revision of the manuscript text and for satisfactory answers to the questions raised by the initial review. I have no further comments.